# Arthroscopic Reduction and Fixation of Coronoid Fractures with Bending K-Wire: A New Technique

**DOI:** 10.3390/jcm11174964

**Published:** 2022-08-24

**Authors:** Chang Hyun Choi, Hyun-Gyu Seok, Sam-Guk Park

**Affiliations:** Department of Orthopedics, Yeungnam University Medical Center, Daegu 42415, Korea

**Keywords:** coronoid process, fracture, elbow arthroscopy, K-wire

## Abstract

The ulnar coronoid process plays a key role in maintaining elbow stability; however, there is no gold standard treatment for ulnar coronoid process fractures. We present a novel surgical technique, arthroscopic reduction and bent K-wire fixation, for type II and III coronoid process fractures according to the O’Driscoll classification. Five patients were treated and retrospectively reviewed between January 2016 and December 2019. All the surgeries were performed by a single surgeon. We evaluated clinical outcomes by evaluating a range of motion, disability of arm, shoulder, and hand (DASH) score, Mayo Elbow Performance score (MEPS), and radiographic images. Intraoperative and postoperative radiographs showed that the fractures healed well and were satisfactorily fixed. The average elbow extension/flexion was −3/130, with an average DASH score of 2.42 and MEPS of 97. Coronoid process fractures can be treated successfully with arthroscopic reduction and fixation of bent K-wire, which allows more rigid fixation and early functional exercise, resulting in good outcomes without special tools.

## 1. Introduction

The ulnar coronoid is a structure known to play a key role in stabilizing the elbow, and recently, there has been growing interest in preserving its function in coronoid process fractures [1]. Ulnar coronoid fractures are commonly accompanied by elbow dislocation and radial head fractures [2]. In an ulnar posterior dislocation cadaver study, Schneeberger et al. demonstrated that if ulnar coronoid process fractures remain unreduced, elbow instability persists, and fixation of coronoid process fractures is important in restoring elbow stability [1,3,4].

Ulnar coronoid process fractures are uncommon, and there is no established gold standard for their surgical treatment. The conventional method of treating ulnar coronoid process fractures is open or arthroscopic reduction and fixation using a suture anchor, 2-0 Ethibond suture, screws, plates, and tension band wiring [1,2,5,6,7,8]. There are various surgical techniques for treating such fractures, but no superior technique has been established [9].

Doornberg et al. demonstrated that type I fractures can be fixed with sutures because they are too small to be fixed with screws, and type II and III fractures are large enough to be fixed with plates and screws [10]. We performed arthroscopic reduction and fixation using bent K-wires on type II and III coronoid fractures and achieved good results. Herein, we discuss arthroscopic reduction and fixation using bent K-wires.

## 2. Surgical Technique

Arthroscopic surgery was performed using general anesthesia or brachial plexus block. After anesthesia, the patient was placed in the decubitus position with the elbow joint on an arm holder 100° flexed and 90° internally rotated. The arthroscope was inserted with the elbow joint flexed at 100° and the forearm dangling freely. A pneumatic tourniquet was positioned high on the upper arm and inflated to 260 mmHg after limb exsanguination. Before creating arthroscopic portals, the elbow joint was expanded by injecting sterile saline (20 mL) through an 18-G needle inserted in the proximal posterior area of the elbow joint. Further, after creating an anterior medial (AM) portal 2 cm proximal to the muscular septum, diagnostic arthroscopy was performed to check for ulnar coronoid process fracture and to clean the hematoma and damaged soft tissue around the bone fragment inside the joint. So as to avoid iatrogenic damage to the brachial artery and median nerve when creating the AM portal, it is recommended to make an incision only on the skin and create the portal using a needle with a cannula. Subsequently, a 30° arthroscope was inserted through the anterior lateral portal, which was created with the forearm pronated to displace the posterior interosseous nerve to the AM side and protect it from iatrogenic lesions. If the intra-articular space is narrow, a 2.7 mm arthroscopy may be helpful. Subsequently, a K-wire (1.5 × 228 mm, 0.045 inch) was inserted from the lateral surface of the ulnar shaft using the ACL drill guide system (RetroConstruction Drill Guide System, Arthrex^®^, Naples, FL, USA) (Figure 1) and passed only to the base of the coronoid process without entering the bone fragment (Figure 2a). The K-wire was then advanced using Kocher (Ochsner) (Figure 2b) or Sponge forceps (Figure 2c) through the bone fragment (Figure 2d). During surgery, proper reduction and the exact position of the K-wire were confirmed using C-arm radiography. After confirming with arthroscopy that the K-wire passing through the bone fragment was long enough to be bent, the tip of the K-wire was bent (Figure 2e), and the coronoid process bone fragment was fixed at the end of the bent part (Figure 2f). The K-wire was then pulled in the direction of the entrance to firmly fix the bone fragment to the ulna (Figure 2g). A C-arm radiograph was used to confirm whether the full range of motion (ROM) of the elbow was appropriate and whether there was a reduction in the stability of the ulna. The K-wire protruding out of the skin was cut, and the tip was bent and buried under the skin.

## 3. Postoperative Care

A posterior semi-plaster bandage was applied to the elbow joint for 2–4 weeks after the surgery, and gradual movement of the elbow joint was permitted from the first day postoperatively. Follow-up radiographic images were obtained four weeks postoperatively. If the bone union was observed, the semi-plaster bandage was removed, and active muscle training exercises were started.

Three months after surgery, sufficient bone union was confirmed, and the K-wire was removed.

## 4. Case Series

The indication for surgery is O’Driscoll classification type II and III ulnar coronoid process fractures that have more than 50% articular invasion. O’Driscoll classification type I ulnar coronoid process fractures were not eligible for inclusion in this study because of the small size of the bone fragments.

We operated on five patients using the above surgical method between 2016 and 2019, and all the surgeries were performed by a single surgeon and retrospectively reviewed by an independent researcher at a mean follow-up period of 23.4 months (13–36 months). The mean patient age was 41.8 years (26–57 years). General anesthesia was used in four patients and brachial plexus block in one patient (Table 1). The ROM of the elbow joint was measured at the last outpatient follow-up. Clinical evaluation was performed at 6 and 12 months at the last follow-up, wherein elbow ROM, disability of arm, shoulder, and hand score (DASH score), and Mayo Elbow Performance score were assessed (Table 2). In order to confirm osseointegration, radiographic evaluation was performed using successive anteroposterior, lateral, and oblique radiographs of the elbow. A typical case is shown in Figure 3.

## 5. Discussion

Various treatment methods for ulnar coronoid process fractures have been reported; however, the best treatment method has not been determined [9,11]. Conservative treatment can be performed if the bone fragment is small; however, surgical treatment is necessary if the bone fragment is large because large fragments cause instability [12].

Open and arthroscopic reduction methods are available. Open reduction can achieve accurate reduction, but it has disadvantages in that it is invasive, causes considerable damage to the surrounding soft tissues, and takes a long time to recover after surgery and reach the functional ROM of the elbow joint. In contrast, the arthroscopic reduction technique is non-invasive and shows a good prognosis; however, it can yield rather poor results if the operator is not skilled with arthroscopic surgery because of the long learning curve [13,14]. Methods for fixing bone fragments for ulnar coronoid process fractures include suture anchors, sutures through drill holes, arthroscopic buttons (Endobutton, Smith&Nephew, London, UK), tension band wiring, screw fixation, or buttress plating [2,15,16]. The method using screws and plates may be advantageous in that it has strong fixation power; however, since it requires an open approach for fixation, relatively long skin incisions are required, a considerably large surrounding soft tissue area is damaged, and subsequent removal surgery may be required, which may also result in soft tissue damage [5,16]. Additionally, suture anchors or thread sutures are suitable for type I coronoid process fractures with small fragments but are not suitable for fractures with large fragments, such as type II and III coronoid process fractures, because their fixation force is relatively weak [17].

In our study, a novel method different from the existing arthroscopic reduction and K-wire fixation technique was used to reduce fractures arthroscopically. In this method, we percutaneously inserted the K-wire, bent the tip, and applied pressure to the bone fragment to increase fixation force.

The fixation method using the bent K-wire technique presented in this study has several advantages. First, a sufficient fixation force can be obtained compared to the suture technique with a suture anchor or Ethibond in the coronoid process fracture of a relatively large bone fragment, which requires sufficient fixation force. Second, compared to fixation with screws and plates that require open reduction, the technique in this study (arthroscopic reduction) causes less damage to the surrounding soft tissue, and the removal surgery is simple [6]. Third, special instruments, such as screws, plates, and suture anchors, are not required [6,18]. Fourth, sufficient fixation force can be obtained, and recovery after surgery is fast; therefore, it is possible to start ROM exercises early, and there is a high possibility of obtaining a functional ROM of the elbow joint later [6]. Fifth, because the K-wire enters the lateral surface of the ulnar shaft, the possibility of damage due to skin friction at the K-wire entry site is low. However, since arthroscopy of the elbow joint is more difficult than other surgical techniques, the disadvantage is that if the operator is not skilled enough with arthroscopic surgery on the elbow joint, it may lead to poor results, such as prolonged operation time.

The limitations of this study are its retrospective design and the small number of participants. Nevertheless, this case study is significant in that it suggests a new surgical option for isolated type II and III ulnar coronoid process fractures, which are rare. In this study, good results of arthroscopic reduction and fixation of the bent K-wire were confirmed in patients with type II and III isolated coronoid process fractures. Therefore, we believe that arthroscopic reduction and fixation of the bent K-wire in patients with isolated type II and III coronoid process fractures provide good functional results as another novel alternative surgical technique.

## Figures and Tables

**Figure 1 jcm-11-04964-f001:**
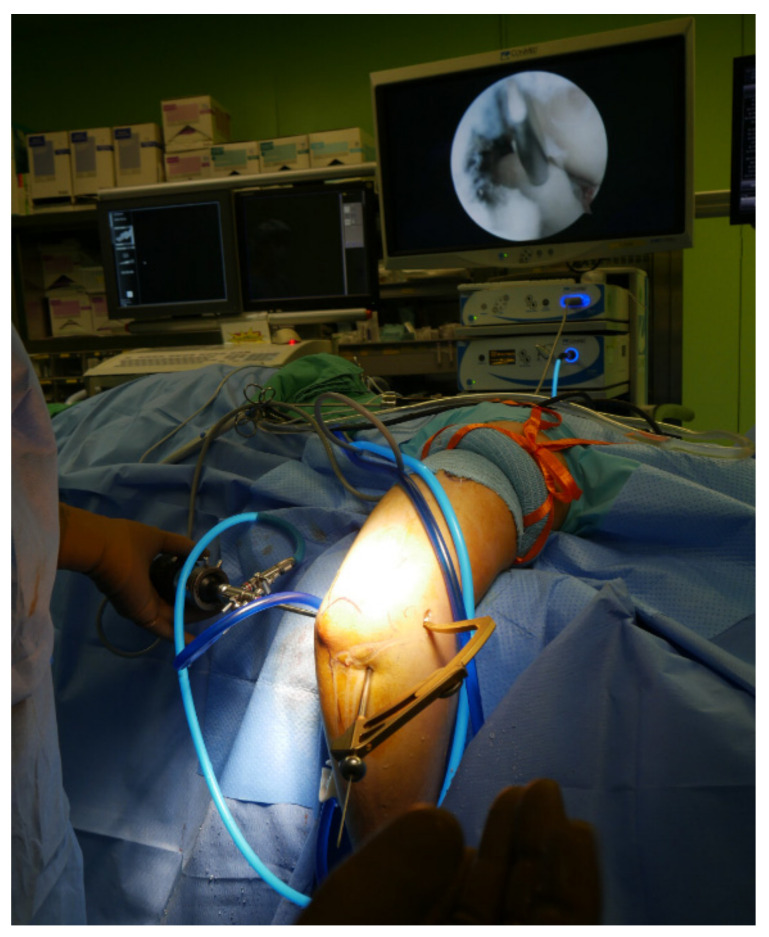
A K-wire (1.5 × 228 mm, 0.045 inch) was inserted from the lateral surface of the ulnar shaft and passed only to the base of the coronoid process without entering the bone fragment using the ACL drill guide system (RetroConstruction Drill Guide System, Arthrex^®^).

**Figure 2 jcm-11-04964-f002:**
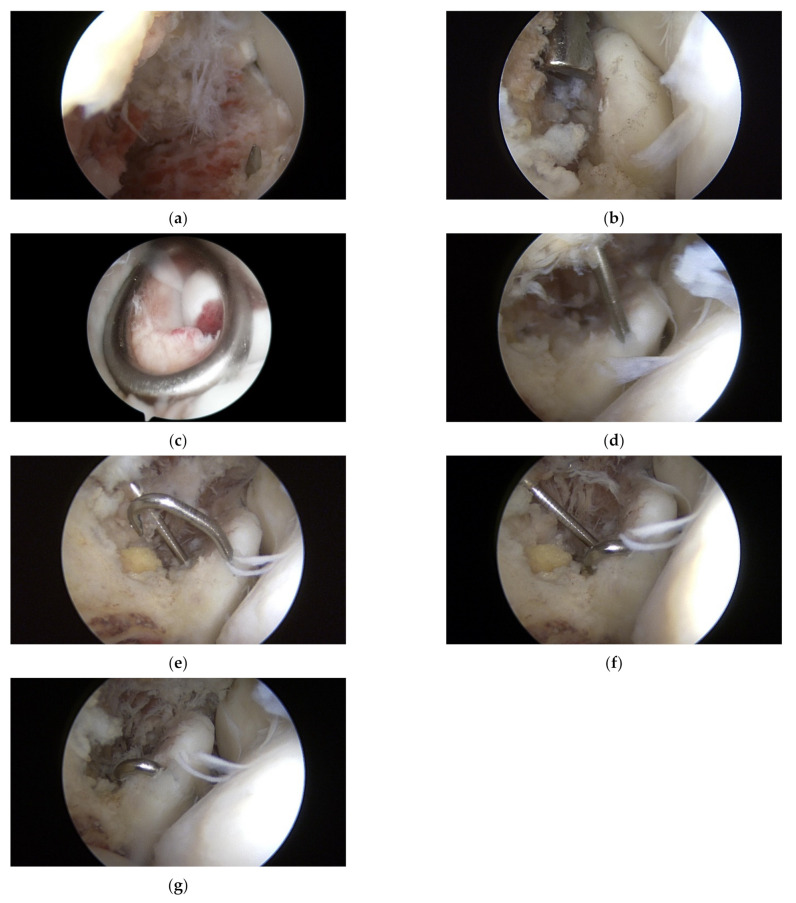
K-wire was inserted from the lateral surface of the ulnar shaft and passed only to the base of the coronoid process without entering the bone fragment (**a**). The K-wire was then advanced using Kocher (Ochsner) (**b**) or Sponge forceps (**c**) through the bone fragment (**d**). After confirming with arthroscopy that the K-wire passing through the bone fragment was long enough to be bent, the tip of the K-wire was bent (**e**), and the coronoid process bone fragment was fixed at the end of the bent part (**f**). The K-wire was then pulled in the direction of the entrance to firmly fix the bone fragment to the ulna (**g**).

**Figure 3 jcm-11-04964-f003:**
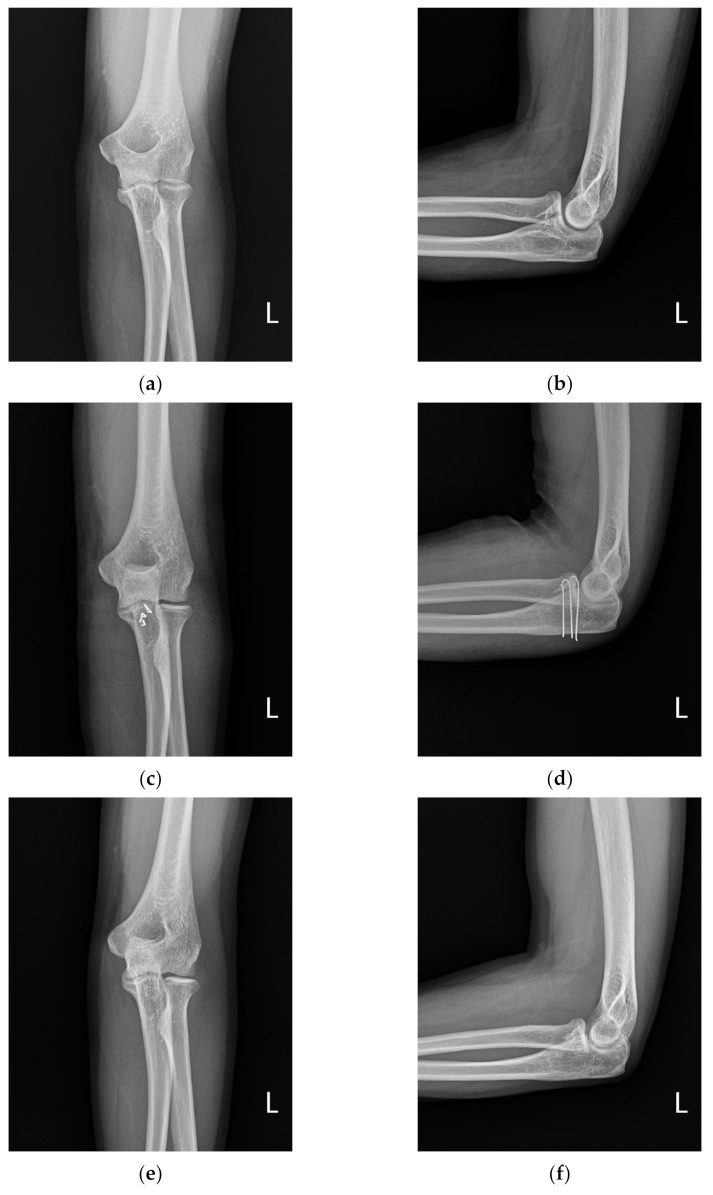
A 32-old-man presented with a coronoid fracture (**a**,**b**); arthroscopic reduction of the coronoid fracture was performed, and fixation was performed with a bending K-wire (**c**,**d**); three months after the operation, the k-wire was removed, and bone union was obtained (**e**,**f**); good function was achieved at 1-year follow-up (**g**–**j**).

**Table 1 jcm-11-04964-t001:** The summary of the case.

Case	Gender	Age	Anesthesia	Injury Mechanism	O’Driscoll Classification	Fragment Size (mm)	Follow Up (Months)	Associated Injury
1	M	29	Regional	Slip down	II (subtype 2)	11 × 21	11	None
2	M	43	General	Slip down	III (subtype 1)	9 × 16	6	None
3	M	54	General	Slip down	III (subtype 1)	11 × 18	12	* MCL, LUCL rupture
4	M	57	General	Fall down	II (subtype2)	7 × 18	12	** Radial head fracture LUCL rupture
5	M	26	General	Fall down	II (subtype1)	8 × 9	18	** Radial head fracture LUCL rupture
Mean		42					12	

* MCL: Medial Collateral Ligament, LUCL: Lateral Ulnar Collateral Ligament. ** For patients with radial head fracture, Open reduction and Internal fixation was performed.

**Table 2 jcm-11-04964-t002:** Clinical outcomes.

Case	Extension (0)	Flexion (150)	Internal Rotation (80)	External Rotation (80)	Range of Motion	MEPS	DASH
1	0	130	85	85	300	100	1.7
2	−10	130	85	80	285	100	0.9
3	0	150	90	90	330	100	0
4	0	120	90	80	290	100	0
5	−5	120	70	50	235	85	9.5
Mean	−3	130	84	77	288	97	2.42

MEPS: Mayo Elbow Performance score. DASH score: Disabilities of Arm, Shoulder, and Hand.

## Data Availability

Not applicable.

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
