# Peer review of "Arthroscopic Reduction and Fixation of Coronoid Fractures with Bending K-Wire: A New Technique"

_jcm, 2022, doi:10.3390/jcm11174964_

Round 1

Reviewer 1 Report

The topic and technique are very good and attractive.

I have a few points and questions;

- Did the patients who were operated on in this way have accompanying injuries or fractures or not? (Terrible triad, elbow dislocation, collateral ligament injury)

- How long after the injury surgery was performed for the patients?

- If possible, explain more about the technique and tricks during the redaction of the fragment

- The method of bending the end of the pin (line 67), with what equipment and technique, and how to take care of the loss of reduction.

- Showing before and after X-ray  and C-ARM images during operation makes the article more attractive

Reviewer 2 Report

Dear Authors, the assessment of your paper is attached. 

Reviewer 3 Report

The article describes a series of patients with coronoid fractures treated with arthroscopic reduction and fixation with bending K-wire.

The topic is extremely interesting and the proposed technique seems to be a valid alternative to ORIF in coronoid fractures with large fragments that cannot be fixed with arthroscopic sutures.

The article is well written. In particular: the introduction is short and complete, the surgical methods and technique are well described, the images are useful and of good quality, the results are clear, the discussion covers all points of interest, and the conclusions are consistent with the results.

Considering the relevance of the topic discussed, the clear contribution to the literature on the topic and the quality of the presentation.

I would only request a small change to the first sentence, removing the word "stability" at line 24. 

Thank you.
